# Assessment of Executive Function in Patients with Traumatic Brain Injury with the Wisconsin Card-Sorting Test

**DOI:** 10.3390/brainsci10100699

**Published:** 2020-10-01

**Authors:** Lizzette Gómez-de-Regil

**Affiliations:** Hospital Regional de Alta Especialidad de la Península de Yucatán Calle 7, No. 433 por 20 y 22, Fraccionamiento Altabrisa Mérida, Yucatán 97130, Mexico; gomezderegil@gmail.com; Tel.: +52-(999)-942-7600; Fax: +52-(999)-254-3535

**Keywords:** executive function, traumatic brain injury, Wisconsin Card-Sorting Test

## Abstract

This review aimed at providing a brief and comprehensive summary of recent research regarding the use of the Wisconsin Card-Sorting Test (WCST) to assess executive function in patients with traumatic brain injury (TBI). A bibliographical search, performed in PubMed, Web of Science, Scopus, Cochrane Library, and PsycInfo, targeted publications from 2010 to 2020, in English or Spanish. Information regarding the studies’ designs, sample features and use of the WCST scores was recorded. An initial search eliciting 387 citations was reduced to 47 relevant papers. The highest proportion of publications came from the United States of America (34.0%) and included adult patients (95.7%). Observational designs were the most frequent (85.1%), the highest proportion being cross-sectional or case series studies. The average time after the occurrence of the TBI ranged from 4 to 62 years in single case studies, and from 6 weeks up to 23.5 years in the studies with more than one patient. Four studies compared groups of patients with TBI according to the severity (mild, moderate and/or severe), and in two cases, the studies compared TBI patients with healthy controls. Randomized control trials were seven in total. The noncomputerized WCST version including 128 cards was the most frequently used (78.7%). Characterization of the clinical profile of participants was the most frequent purpose (34.0%). The WCST is a common measure of executive function in patients with TBI. Although shorter and/or computerized versions are available, the original WCST with 128 cards is still used most often. The WCST is a useful tool for research and clinical purposes, yet a common practice is to report only one or a few of the possible scores, which prevents further valid comparisons across studies. Results might be useful to professionals in the clinical and research fields to guide them in assessment planning and proper interpretation of the WCST scores.

## 1. Introduction

Traumatic brain injury (TBI) is an alteration in normal brain function or any other evidence of brain pathology caused by an impact from external mechanical forces, such as rapid acceleration or deceleration, a bump or jolt to the head or penetration by a projectile. About sixty-nine million people experience TBI from all causes each year, with the Southeast Asian and Western Pacific regions experiencing the greatest overall burden of disease [1]. TBI is mild, moderate or severe, depending on the resulting severity and duration of loss of consciousness, post-traumatic amnesia and neuro-radiological evidence of cerebral damage. This classification system is highly reliable for first diagnosis; however, its prognostic value for long-term neuropsychological outcome is still limited as it rarely takes into account premorbid factors, underlying structural damage and the impact of non-neurological factors [2].

The main outcomes following TBI include mortality, functional disability, health-related quality of life, and cognitive, psychiatric and social complications [3]. Anatomically, frontal lobes are particularly vulnerable to TBI, given their site at the front of the brain and their large size and because of scraping of the orbitofrontal region against the fosse. Damage to the prefrontal cortex is highly, although not exclusively, associated with impairment of executive function, a complex set of abilities that include various skills such as working memory, initiating or inhibiting behavior, cognitive flexibility and decision-making, as well as planning and organization, monitoring performance, problem-solving, metacognition, learning rules, controlling emotions, multitasking, self-awareness, social behavior and motivation [4,5,6]. Patients with TBI may experience only some of these difficulties, and further, these difficulties might be almost imperceptible. Unawareness of executive dysfunction is common in patients with TBI, and signals might be only noticeable to those people in close and frequent interaction with the patient. On the other hand, other symptoms might be so intrusive that they can affect negatively the patient’s daily activities, emotional wellbeing and social interactions.

Neuropsychological assessment of TBI often occurs during inpatient hospitalization if post-traumatic amnesia has resolved [7]. Executive dysfunction, especially if mild, might not be noticeable immediately after experiencing a TBI, but rather manifest later on when the individual starts reintegration to his/her usual activities. Among various tests to assess executive function, the Wisconsin Card-Sorting Test (WCST) is one of the most widely known and used in clinical and research practice. Laxe and colleagues, in a systematic review of scales used in TBI patients, found that 15% of 193 selected research articles included the WCST [8] as a measure.

Grant and Berg developed the WCST in 1948, but it was not until 1963 that Milner introduced it as a tool to assess prefrontal lobe dysfunction in patients with brain lesions. The original WCST presents 4 key cards and 128 response cards with geometric figures (e.g., squares, triangles, circles and stars) varying according to color, form or number. A shorter version, the WCST-64, includes only 64 cards. The interviewer presents the stimulus cards one by one, and the person must select a card matching the stimulus according to a criterion in turn. The matching criterion can be a color, figure or number. After each matching, the interviewer tells the interviewee if the answer was correct or not, and learning must follow through trial and error. The classification criterion changes after the interviewee yields 10 consecutive correct matches without warning, demanding a flexible cognitive shift. There is no time limit and the test finishes when the individual sorts all the cards or if he/she achieves six correct sorting criteria. Once completed, the total and percentage of the following scores are calculated: number of errors, perseverative responses, perseverative errors, nonperseverative errors and conceptual level responses. The manual provides tables, according to age and/or years of education, to convert raw scores into standard scores, T scores and percentile scores. Additionally, the number of categories completed, trials to complete first category, failure to maintain set (when an error follows five or more correct consecutive answers) and learning to learn scores can be calculated and transformed into percentiles. From these various scores, the numbers of perseverative errors, nonperseverative errors and categories completed are the most commonly used for reporting the patient’s performance (see [9] for a review and update).

Frontal lobes, responsible for the executive function, are likely to be damaged after TBI. Executive function plays a significant role in the daily performance of the person, and due to its close connection to other cognitive functions, reliable measures of executive function are of importance for clinical and research purposes. Although not the only one, the WCST is one of the most widely known and used instruments to assess executive function in a variety of clinical populations with brain dysfunctions. Nevertheless, and to the best knowledge of the author, no publication is yet available that summarizes research using this tool in patients with TBI. This information, properly organized, might be useful to professionals in the clinical and research fields to guide them in assessment planning and proper interpretation of results. This review aims at providing a brief and comprehensive summary of recent research making use of the WCST to assess executive function in patients with TBI.

## 2. Methods

The author performed a bibliographical search in the PubMed, Web of Science, Scopus, Cochrane Library and PsycInfo databases. The terms “traumatic brain injury” and “TBI” were entered along with “Wisconsin card-sorting test” and “WCST”. Inclusion criteria were: (1) research papers, (2) published in peer-reviewed journals, (3) published during the last decade (2010 to May 2020) and (4) available in English or Spanish. Exclusion criteria were: (1) not original research (e.g., letters, dissertations, reviews and/or meta-analyses) and (2) content not related to the objective of the study (i.e., study design not including patients with TBI and/or not using the WCST). The author accessed the online resources on 20 May 2020. After applying the inclusion and exclusion criteria, a final list of references was generated and the full content of the manuscripts was consulted to verify their relevance to the objectives of the review. Through complete and thorough readings of the manuscripts, the following data were collected: year of publication, research team setting, participant age stage (pediatric or adult) and study design. Regarding participants, the sample size, age, gender, severity of TBI and time since TBI were recorded. Concerning the WCST, the version used and reported scores were registered, along with its use in the study.

## 3. Results

An initial search produced 387 results from the five sources, which was reduced to 186 by filtering duplicated results. After applying the exclusion criteria and reviewing abstracts and/or manuscripts, the list was reduced to 47 relevant publications (Figure 1).

Table 1 summarizes the basic features of the publications. The highest proportion of publications came from the United States of America (34.0%), followed by Italy (14.9%) and other countries worldwide (51.1%). Regardless of the language spoken in the setting of research, all publications provided an abstract written in English, yet two manuscripts were in Spanish. The highest proportion of publications (*n* = 20) came from settings whose official language is English (United States of America, Canada and Australia), and the remainder (*n* = 27) came from countries with diverse languages, such as Italian, Spanish, Portuguese, French, Hebrew, Japanese, Korean, Malay, Serbian and Thai. Most studies included adult samples (95.7%), and only two (4.3%) had pediatric samples.

Table 2 summarizes information about the design of the studies with the corresponding references. Observational (descriptive) designs were the most frequent (85.1%), particularly cross-sectional and case series studies. There were seven randomized control trials. The original WCST version, comprising 128 cards and applied by an interviewer, was the most frequently used version (78.7%). Only three studies (5.4%) made use of computerized versions. Regarding WCST scores, perseverative errors (46.8%) followed by categories completed (42.6%) and perseverative responses (31.9%) stood out as the most often reported. In the selected studies, the use of the WCST was diverse. Characterization of the clinical profile of participants was the most frequent purpose (34.0%), in accordance with the high percentage of observational studies. In some cases (38.3%), WCST scores aimed at detecting differences in the executive function performance of TBI patients in comparison to clinical groups or healthy controls, or between groups of TBI patients defined by a selected clinical criterion (e.g., self-awareness, history of suicide attempt, anosmia).

Regarding the characteristics of the TBI patients participating in the studies, in some cases, samples included mostly (57.4%) or only (6.4%) men, and others included mostly (8.5%) or only (10.6%) women. Sex was not reported in eight (17.0%) manuscripts. Mean age in pediatric samples ranged from 14 to 15 years and from 20 to 77 in adult samples. The average time after the occurrence of the TBI, as reported in 34 manuscripts, ranged from 4 to 62 years in single case studies, and from 6 weeks up to 23.5 years in the studies with more than one patient. Although not stated, the time intervals suggest that all patients had passed the acute phase. Severity of the TBI in participants was reported in 37 manuscripts. Most studies (23.4%) included patients with a mild to severe TBI, followed by samples with moderate to severe TBI (12.8%) and mild to moderate TBI (10.6%). Some studies included exclusively patients with mild (14.9%), moderate (2.1%) or severe (14.9%) TBI.

A total of 32 studies exclusively included patients with TBI: six single cases, 11 analyzing all participants as a single group, eight grouping participants according to some clinical feature (e.g., severity of TBI, disability, history of suicide attempt, anosmia) and seven randomized control trials comparing groups by treatment (e.g., cognitive rehabilitation, growth hormone replacement therapy, sertraline medication, neuro-feedback training). On the other hand, 15 studies included not only TBI patients but also healthy controls (*n* = 12), clinical controls (*n* = 2) or both a healthy control group and clinical control group (*n* = 1). Only four studies compared the performance of patients according to severity (mild, moderate and/or severe); two of them also including a healthy control group. Most studies (72.3%) used the WCST scores as a descriptive feature of participants, comparing between groups or not. Table 3 presents the results and references.

Additionally, in an effort to offer an overview of the performances across studies, the author selected the ones using the 128-card version, the most often used, and reporting perseverative errors and/or completed categories scores, which were the most often available. Table 4 summarizes the results. Compared to healthy controls, TBI patients produced more perseverative errors, as did those with a severe TBI in comparison to those with a moderate or mild TBI. Given the differences across samples and the type of reported scores, further analyses were not possible. Regarding the number of completed categories (possible range: 0 to 6), in a couple of studies, participants scored the highest, while most scores ranged from 4 to 5. In addition, some scores from patients with TBI were not so distant from those from healthy controls.

Only 12 publications [12,22,25,30,38,41,42,43,46,47,50,51] reported including effort measures to account for performance/symptom validity. The preferred measures were the Trail Making Test (e.g., total completion time in seconds), the Computerized Assessment Response Bias and the California Verbal Learning Test (e.g., trials 1–5 learning score, long-delay free-recall score).

The author and an invited rater independently assessed the quality of the six randomized control trials with the corresponding Critical Appraisal Skills Programme (CASP) checklist [57] (Table A1 in Appendix A). Interrater reliability by intraclass correlation coefficient (by two-way mixed model and absolute agreement) was 0.70 (95% CI = 0.28–0.67).

## 4. Discussion

Here, the author presents a review providing a brief and comprehensive summary of recent research using the WCST to assess executive function in patients with TBI. The United States of America has produced most of the studies reviewed, yet they represent research from 17 different countries, from all continents. Worldwide, TBI is recognized as a health priority, given its related global burden (estimated in 2016 as 8.1 million years of life lived with disability; YDLs) and an expected increment in its incidence in view of the exponential population growth, population ageing and more frequent use of vehicles [58]. The WCST is not language-based; instructions are simple and oral interaction between the interviewer and the interviewee can be minimal. This is an important feature in applying the WCST across settings with different languages, yet always keeping in mind the need for standardized scores.

Reports from pediatric samples are scarce. This might be due not only to the limited age range compared with adults, but also to the fact that adults are more likely to be exposed to environmental conditions (e.g., traffic collisions, combat injuries) that may lead to a TBI. Moreover, it must be kept in mind that younger children (i.e., infants and toddlers) are also at very high risk of TBI; however, the WCST is not meant for this population, and other available measures should be considered (e.g., the Behavior Rating Inventory of Executive Function—Preschool Version (BREF-P) and the Developmental NEuroPSYchological Assessment (NEPSY)). The WCST has been standardized for people aged from six and a half to 89 years of age. Demographically corrected normative data provides score profiles according to age: 13 standardized scores tables for children/adolescents (6.5 to 20 years old) and 60 standardized scores tables for adults (21 to 89 years old), which also consider the number of years of education. This does not mean that the WCST is rarely applied to the pediatric population, as it is often used for other clinical conditions which are more common in infants in comparison to TBI, such as attention deficit hyperactivity disorder (ADHD) and autism [59].

The original and most widely used version of the WCST includes 128 cards to match. However, a shortened version presenting only the first deck of 64 cards is available, particularly for assessment situations with time restrictions or when the patient’s attention span is compromised, as might be the case with elderly [60]. There is also the Modified WCST (M-WCST), a shorter, simpler and less ambiguous version that includes two sets of 24 cards. The M-WCST lacks response cards which share more than one feature with the stimulus cards, the interviewee’s first response sets the category criteria, and it estimates perseverative errors differently [61]. More recently, computerized versions have become available, which have the advantages of a more efficient use of resources, improving reliability by equal assessments, and decreasing errors in test presentation, response recording and scoring. This could be very useful for research purposes, particularly when recruiting big samples. Yet, scores from the computerized versions do not seem quite equal to those of manual versions, and new norms for computer versions still need to be established [62].

As an instrument to assess executive function in TBI, the WCST scores have served different purposes according to the studies’ designs. In most cases, scores were included as an index of the TBI participants’ clinical status, in observational studies with a single case, case series or comparison between groups. Results from these studies helped to gather evidence of the sensitivity (potential to detect those with executive dysfunction) and specificity (potential to exclude those with no executive dysfunction) of the WCST, being essential features of diagnostic instruments. Although for research purposes, using the WCST scores as an index of executive dysfunction might be practical when assessing an individual in clinical practice, relying on a single indicator might lead to invalid conclusions. The WCST should be part of a larger and comprehensive neuropsychological battery, and its results should be interpreted along with information from different sources, such as interviews with the patient and relatives, observations and behavioral checklists. Evidence has shown that the capacity of performance-based executive measures, such as the WCST, to predict a patient’s ability to function adaptively in daily life after TBI is variable. Patients with compromised executive function can perform well on tests of executive functioning, but demonstrate real-world behavioral disturbances with a reduction of autonomy. Thus, clinicians and researchers must consider the ecological validity of the selected measures and complement them with behavioral data from other sources Kibby and colleagues [63], in testing the ecological validity of the WCST, found that perseverative responses did not significantly correlate with level of job performance, yet it did predict occupational status (from manual labor to higher-level positions). On their part, Pezzuti and colleagues [64] constructed and validated an ecological version of the WCST aimed at the elderly, that was found to be more discriminating and to have more advantages than the traditional versions. The ecological value of the WCST for discriminating functional outcomes, rather than just evidencing the severity of the injury or the presence of a condition, comes forward as a topic worthy of further research.

Randomized control trials used the WCST scores as an outcome variable when testing the effectiveness of a medical treatment (e.g., sertraline [54], growth hormone therapy [52]) or a rehabilitation program (e.g., CogSmart [50,51]; STEP [53]; neuro-feedback [55], vocational problem-solving [56]). The scores obtained before group allocation and treatment, as in the case of descriptive studies, might be useful to evidence the WCST’s sensitivity and specificity.

Once the person completes the WCST, all scores can be calculated, yet the studies only reported one or a few of them. Perseverative errors were the scores most often reported. A perseverative response occurs when the interviewee matches a card using the same criterion (color, form or number) used in the immediate previous match, regardless of the response being correct or not. Thus, perseverative errors refer to the number of perseverative responses that were not correct; i.e., the response did not match the valid criterion in turn. Perseverative errors index an incapacity to inhibit a learned response despite knowing from feedback that the response is incorrect. As a measure of executive function, we expect that the greater the dysfunction, the higher the score in perseverative errors. The reported results seem to agree with this, yet the diversity in samples prevents valid comparisons to decide whether there is a significant association between TBI severity and executive dysfunction. Another often-reported score is the number of categories completed; that is, the number (0 to 6) of categories with a sequence of 10 consecutive correct matches to the criterion in turn. Once the subject completes a category, the sorting criterion changes. The test ends when the person completes all six categories or after he/she sorts all the stimulus cards. The ability for completing categories represents the person’s capacity to promptly identify the sorting criterion and persevere in providing the correct responses. In some cases, the studies reported that patients with TBI achieved the highest score, while in other cases their scores were not significantly low, and even not so different from those of the controls. This may suggest that patients with TBI have a good ability to figure out the sorting criterion, properly adapting their responses from feedback, despite having difficulties in promptly shifting the criterion, as higher scores in perseverative errors suggest. Raw scores from the interview can be transformed to T scores and percentile scores, which would be very useful for research and clinical purposes, respectively. It would be ideal if all WCST scores were available from the manuscripts, so that further analyses with data across studies could be made, providing a valid insight into the executive function profile of patients with TBI.

The only author performed the bibliographical search, data collection and analyses, which may be a source of bias. In addition, due to language limitations, four manuscripts that might be relevant to the study’s objective (two in Chinese, one in French and one in Turkish) could not be included.

## 5. Conclusions

This brief review of recent research on the use of the WCST to assess executive function in patients with TBI showed that interest in the topic is worldwide and has been mainly focused on adult populations. Although shorter and/or computerized versions are available, the original WCST with 128 cards is still the most often used. The WCST is a useful tool for research and clinical purposes, yet a common practice is to report only one or a few of the possible scores, which prevents further valid comparisons across studies.

## Figures and Tables

**Figure 1 brainsci-10-00699-f001:**
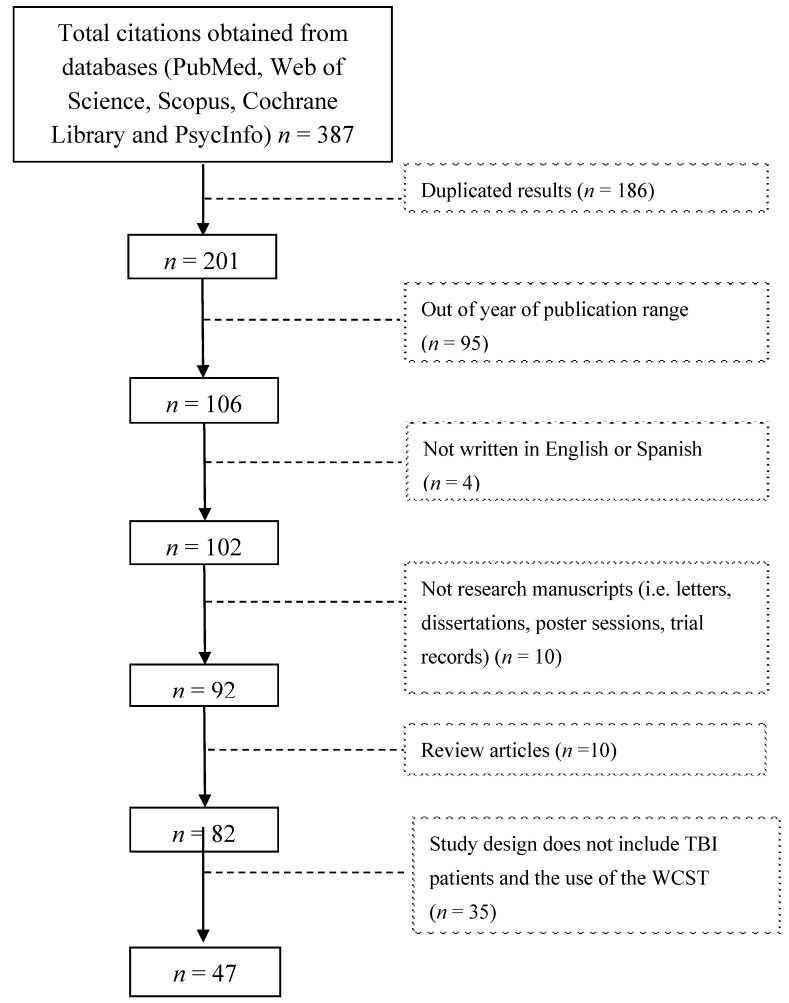
Study flow diagram.

**Table 1 brainsci-10-00699-t001:** Basic features of publications (*n* = 47).

Year of Publication: *n* (%)					
2010:	5 (10.6)	2011:	9 (19.1)	2012:	6 (12.8)
2013:	5 (10.6)	2014:	5 (10.6)	2015:	4 (8.5)
2016:	4 (8.5)	2017:	3 (6.4)	2018:	2 (4.3)
2019:	3 (6.4)	2020:	1 (2.1)		
**Location: *n* (%)**					
Australia	2 (4.3)	India	2 (4.3)	Serbia	1 (2.1)
Brazil	3 (6.4)	Italy	7 (14.9)	South Africa	1 (2.1)
Canada	2 (4.3)	Israel	2 (4.3)	Spain	2 (4.3)
China	2 (4.3)	Japan	1 (2.1)	Thailand	1 (2.1)
Colombia	1 (2.1)	Korea	1 (2.1)	United States of America	16 (34.0)
France	2 (4.3)	Malaysia	1 (2.1)		

**Table 2 brainsci-10-00699-t002:** Study design features of publications (*n* = 47).

**Study Design: *n* (%)**		**References**
**Observational**		
Case report	6 (12.8)	[10,11,12,13,14,15]
Case series	10 (21.3)	[16,17,18,19,20,21,22,23,24,25]
Cohort	1 (2.1)	[26]
Cross-sectional one group	10 (21.3)	[27,28,29,30,31,32,33,34,35,36]
Cross-sectional two or more groups	13 (27.7)	[37,38,39,40,41,42,43,44,45,46,47,48,49]
Experimental		
Randomized control trial	7 (14.9)	[50,51,52,53,54,55,56]
**WCST version: *n* (%)**		**References**
With interviewer		
WCST-128	37 (78.7)	[10,11,12,13,14,15,16,17,18,19,20,21,24,25,27,28,29,31,32,33,34,35,36,37,38,39,40,42,43,45,46,47,48,49,52,55,56]
WCST-64	6 (12.8)	[26,30,41,50,51,54]
WCST-48	1 (2.1)	[44]
Computerized		
WCST-128	2 (4.3)	[23,53]
WCST-64	1 (2.1)	[22]
**WCST reported scores: *n* (%)**		**References**
Correct responses	5 (10.6)	[16,23,26,37,52]
Errors	13 (27.7)	[10,15,16,19,23,31,34,37,41,42,44,46,56]
Perseverative responses	15 (31.9)	[10,16,17,20,23,30,31,32,34,36,37,38,42,45,55]
Perseverative errors	22 (46.8)	[14,15,16,23,24,25,26,27,28,31,33,37,39,41,44,45,47,49,50,51,54,56]
Nonperseverative errors	11 (23.4)	[15,16,23,28,31,37,39,42,44,45,49]
Conceptual-level responses	9 (19.1)	[10,15,16,31,34,42,54,55,56]
Categories completed	20 (42.6)	[14,15,16,17,20,23,24,31,32,33,34,36,37,41,42,43,44,45,46,49]
Trials to complete first category	6 (12.8)	[16,23,34,42,43,46]
Failure to maintain set	5 (10.6)	[31,34,42,46,47]
Learning to learn	0 (0)	
**Use of WCST scores in the study: *n* (%)**		**References**
Description of clinical profile of participants	16 (34.0)	[10,12,13,14,15,16,17,19,20,21,22,24,25,27,28,37]
Comparison of clinical profile within groups	18 (38.3)	[29,31,32,33,34,35,36,39,40,41,42,43,44,45,46,47,48,49]
Treatment outcome	7 (14.9)	[26,50,51,52,54,55,56]
Predictor variable	4 (8.5)	[11,18,30,38]
Predicted outcome	1 (2.1)	[23]
Inclusion criterion	1 (2.1)	[53]

**Table 3 brainsci-10-00699-t003:** Characteristics of samples (*n* = 47).

Reference	% of Male Participants	Mean Age (Years) of Patients with TBI	Mean Time after Injury	Severity of TBI	Participants
[10]	0	77	NR	NR	1 TBI patient 1 healthy control
[11]	78	36.26	11.20 months	NR	80 TBI patients
[12]	100	45	NR	mild	1 TBI patient
[13]	100	21	4 years	NR	1 TBI patient
[14]	0	67	62 years	NR	1 TBI patient
[15]	0	20	8 years	severe	1 TBI patient
[16]	76	34	NR	mild to severe	30 TBI patients
[17]	75	30.73	7.04 months	moderate to severe	32 TBI patients
[18]	45	NR	78 of days (median)	NR	69 TBI patients
[19]	0	22	10 years	severe	1 TBI patient
[20]	68	15.1	23.0 months	NR	65 TBI patients
[21]	35	34.5	1671.3 days	mild to severe	19 TBI patients
[22]	87	35.52	NR	NR	94 TBI patients
[23]	NR	43.68	103 months	severe	25 TBI patients
[24]	74	32	8.7 months	moderate to severe	43 TBI patients
[25]	100	35.3	6.6 years	mild	56 TBI patients
[26]	95	25.79	6 weeks	mild to moderate	19 TBI patients 14 healthy controls
[27]	80	29.5	NR	mild	60 TBI patients 32 healthy controls
[28]	NR	35.4	39.1 months	mild	10 nonaphasic speaker TBI patients 13 neurologically intact controls
[29]	57	32	25 months	mild to severe	7 TBI patients 7 patients with right brain damage
[30]	82	37.3	NR	mild to severe	377 TBI patients
[31]	90	30.3	2.13 months	mild	30 TBI patients 30 healthy controls
[32]	91	33.6	6.72 years	severe	11 TBI patients 11 healthy controls
[33]	NR	36.18	48.6 months	moderate to severe	20 severe TBI patients 20 moderate TBI patients 20 healthy controls
[34]	69	25.53	8.6 months	severe	29 TBI patients 38 healthy controls
[35]	85	40.5	335 days (median)	mild to severe	27 TBI patients 18 healthy controls
[36]	80	36.9	262 days	severe	20 TBI patients with adequate levels of self-awareness 20 healthy controls
[37]	0	14.0	<6 months, 7 patients ≥6 months and <12 months, 3 patients ≥12 months, 10 patients	NR	20 TBI patients 7 healthy controls
[38]	45	44.55	NR	mild	95 TBI patients who passed the performance validity measures 60 TBI patients who failed the performance validity measures
[42]	75	37.6	37.7 months	mild to severe	176 TBI patients 49 patients with diffuse neurological impairment 20 healthy controls
[43]	51	51.25	NR	moderate to severe	73 TBI patients 60 patients with no TBI
[44]	77	36.56	22 months	mild to severe	39 mild TBI patients 57 severe TBI patients
[45]	85	30.6	0.9 years median interval	severe	29 TBI patients with good metacognitive self-awareness 23 TBI patients with heightened metacognitive self-awareness
[46]	NR	40.97	954.57 days	mild	77 TBI patients grouped as disabled or impaired
[47]	94	51.2	23.5 years	mild to severe	18 TBI patients with a history of at least one suicide attempt 29 TBI patients with no history of suicide attempt
[48]	90	39.3	NR	mild to moderate	9 mild TBI patients 11 moderate TBI patients 27 healthy controls
[49]	NR	38.18	NR	mild to severe	30 mild TBI patients 30 moderate TBI patients 30 severe TBI patients
[39]	NR	36.6	22.4 months	moderate	10 TBI patients 20 healthy controls
[40]	67	38.8	153 months	moderate to severe	6 planner TBI patients 11 avoider TBI patients
[41]	91	43.29	NR	NR	30 anosmic TBI patients 36 nonanosmic TBI patients
[50]	94	32	4.35 years	mild to moderate	16 TBI patients treated with supported employment plus Cognitive Symptom Management and Rehabilitation Therapy (CogSMART) 18 TBI patients treated with enhanced supported employment
[51]	96	31.76	4.56 years	mild to moderate	25 TBI patients treated with supported employment plus Cognitive Symptom Management and Rehabilitation Therapy (CogSMART) 25 TBI patients treated with enhanced supported employment
[52]	NR	37.6	8.05 years	NR	12 TBI treated with growth hormone replacement therapy 11 TBI treated with placebo
[53]	38	45.3	12.6 years	mild to severe	49 TBI patients treated with a problem-solving and emotional regulation program 49 TBI patients in waitlist
[54]	73	34.9	20.35 days	moderate to severe	49 TBI patients treated with sertraline 50 mg 50 TBI patients treated with placebo
[55]	90	29.54	NR	mild to severe	30 TBI patients treated with neuro-feedback training 30 TBI patients in waitlist
[56]	NR	NR	NR	mild to moderate	20 TBI patients treated with an artificial intelligence virtual reality-based vocational training system 20 TBI patients treated with a psycho-educational vocational training program

NR: not reported; Percent of male participants is rounded to integer numbers.

**Table 4 brainsci-10-00699-t004:** Mean scores for Perseverative Errors and Categories Completed in Studies Using the WCST-128.

Perseverative Errors: Number of Incorrect Matches in Sequence, Following the Same Incorrect Criterion					
[14]	6	[27]	22.10 (mild TBI) 16.34 (healthy)	[47]	41.7 (mild to severe TBI)
[15]	79 (severe TBI)	[28]	8.4 (mild TBI) 0.08 (neurologically intact)	[49]	6.39 (mild TBI) 26.00 (severe TBI) 18.96 (moderate TBI)
[16]	30 (mild TBI) 30 (severe TBI) 11 (moderate TBI)	[31]	19.76 (mild TBI) 15.67 (healthy)	[39]	6.4 (moderate TBI) 0.5 (healthy)
[23]	19.83 (severe TBI)	[33]	7.40 (moderate TBI) 0.06 (healthy) 31.67 (severe TBI)	[56]	31.32 virtual reality group at baseline 31.60 psychoeducation group at baseline
[24]	15.6 (moderate to severe TBI)	[37]	89.42 raw scores		(mild to moderate TBI)
[25]	10.8 (mild TBI)	[45]	14.1 (severe TBI)		
**Categories Completed: Number of categories with a sequence of 10 consecutive correct matches**					
[14]	6	[24]	5 (moderate to severe)	[37]	4.07
[15]	6 (severe TBI)	[31]	4.07 (mild TBI) 4.53 (healthy)	[42]	4.85 controls 4.90/4.12 (mild TBI, good/poor effort) 4.57/4.38 (moderate to severe TBI, good/poor effort) 2.55 diffuse neurological impairment
[16]	3 (mild TBI) 4 (severe TBI) 5 (moderate TBI)	[32]	5.54 (severe TBI)	[43]	3.6/3.6 (moderate to severe TBI, suicide attempt: no/yes) 3.9/4.1 (healthy, suicide attempt: no/yes)
[17]	4.5 (moderate to severe TBI)	[33]	5.35 (moderate TBI) 6.00 (healthy) 3.00 (severe TBI)	[45]	5.1 (severe TBI)
[20]	4.77	[34]	5.17(severe TBI) 5.8 (healthy)	[46]	Weighted mean = 0.354 (mild TBI)
[23]	3.04 (severe TBI)	[36]	4.85 (severe TBI) 5.65 (healthy)	[49]	4.95 (mild TBI) 2.50 (severe TBI) 4.00 (moderate TBI)

## Data Availability

The dataset generated and/or analyzed during the current study is available from the corresponding author on reasonable request.

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
