# Peer review of "Assessment of Executive Function in Patients with Traumatic Brain Injury with the Wisconsin Card-Sorting Test"

_brainsci, 2020, doi:10.3390/brainsci10100699_

Round 1

Reviewer 1 Report

The author appears to have responded comprehensively to the comments and suggestions from both reviewers.

Minor points: add info about clinical implications to the abstract.

Denote which specific metrics from TMT and CVLT were used to measure effort. 

The paragraph on effort measures should be separated from the paragraph describing the CASP.

Reviewer 2 Report

The authors have nicely addressed my comments.  

Author Response

This manuscript is a resubmission of an earlier submission. The following is a list of the peer review reports and author responses from that submission.

Round 1

Reviewer 1 Report

This well-written review examines evidence for use of the WCST in evaluation of executive function after TBI.  It makes an important contribution to the literature.  However, more details on review methodology, and rationale for this methodology, are required.  I would like to see more on clinical implications: why is this "brief and comprehensive summary of recent research regarding the use of WCST in assess EF in patients with TBI" important?  How will this impact care/treatment?  Please also add a data point and section assessing whether each study included effort measures, as that impacts the validity of the WCST scores presented. Minor revisions suggested below.

Abstract: Too much methodological detail, but need date ranges of studies searched.  More rationale for the study needed- is this the first review of WCST for TBI patients?

Introduction: good review of definitions of TBI, EF, and description of the WCST.  Perhaps could be shortened a bit, and more citations are needed.  Minor point: frontal lobes are vulnerable to TBI additionally because of scraping of the orbitofrontal region against the fosse.

Methods: What is a "bibliographical" (line 82)?

More detail needed here- see above.

Please define "thoughtful reading of the manuscripts".  More information on review methodology and criteria for determining study quality needed.

Figure 1 is nice.

Some  throughout the manuscript should be checked for clarity by an English language editing service: For example, lines 17-18 (abstract); lines 44-48, 52-54, 56, and 60-61(intro).

Results: Table 1 is not clear.  Years need to be separated from the n.  Entries in this table need to be justified and organized in a way that increases readability.

Table 2: Nice, informative.  Please justify entries.

Table 3: Very well-organized and informative.

Table 4: Nice addition.  A description of the analyses used to produce these results is required.

Discussion: Well-written.  Don't need the first 3 summarizing sentences; just go right into discussing the results.  Section about lack of studies in pediatric samples doesn't make sense.  Kids  0-4 are at very high risk of TBI but can't be assessed with WCST.  Also clinicians may choose to use other EF measures for pediatric patients and this should be considered.

Conclusions: What is the basis for the statement that "the WCST results must be interpreted in the context of a comprehensive assessment..."  This is true, but I'm not sure how that is a conclusion of this study. 

Reviewer 2 Report

This paper presents a review of the literature on the use of the Wisconsin Card-Sorting Test (WCST) to assess executive function after TBI.  Overall, this is a well-written paper, though minor language editing, especially in the Discussion section, would improve overall quality.

Specific comments are included below:

Abstract:  The authors state that most publications came from the U.S., but the percentage was only 34.0%, which is not “most” (which suggests the majority).  Consider modifying “The highest proportion of publications came from the United States…”  Including a statement in the abstract about whether the studies look at acute vs chronic TBI and injury severity level would also be helpful.

Introduction: Overall, the Introduction is thorough and well-written.  Below are minor edits. The only more substantial comment I have is that the justification for the need for this review remains unclear. What gap will providing this review address? What question is the author trying to answer or to what end is this review being done?  

  • Line 30: change “suffer” to “experience”
  • Line 51: change “suffering a TBI” to “experiencing a TBI”
  • Line 72: change “subject” to “patient” or “individual”
  • Line 74: change “likely to suffer damage because of a TBI” to “likely to be damaged after TBI”
  • Lines 75-76: this is an awkwardly worded and confusing sentence. To whom or what does “their” refer to? (their significant connection)

Methods: Was everything conducted by a single author or was there overlap in evaluating and determining eligibility of manuscripts and verifying data extraction?  If all completed by a single author, the potential biases this introduces needs to be included in the limitations.

Results:

  • Left-justifying content of the tables would make them much more legible.
  • How many articles were in English vs Spanish? And what languages were represented in the original studies (especially those conducted in countries where neither English nor Spanish are the primary languages spoken)?

Discussion:

  • The author rightfully notes that, clinically, the WCST should be used in the context of a more comprehensive assessment. In the same vein, a brief mention of the ecological validity of the WCST – and if this review can speak at all to its ecological validity – would improve clinical utility of this review.  That is, how good is the WCST at predicting or differentiating different functional outcomes rather than just severity of injury or presence of a condition or not.
  • The international perspective of this review is a strength, but a criticism of many neuropsychological tests is that the norm-based scores were derived from the English-language version of the test. This maybe somewhat less of a concern where the WCST is not as verbally-based as many neuropsych tests, but this still is an important point to acknowledge and discuss.